# Evaluation of the Electrocardiographic Tp-e, Tp-e/QT, and Tp-e/QTc Parameters in Patients with Non-Alcoholic Liver Disease

**DOI:** 10.3390/medicina61040766

**Published:** 2025-04-21

**Authors:** Kader Eliz Sahin, Mesut Karatas, Sezgin Barutcu, Ibrahim Halil Inanc

**Affiliations:** 1Department of Cardiology, Kocaeli City Hospital, Kocaeli 41060, Turkey; 2Department of Cardiology, Kosuyolu High Specialization Education and Research Hospital, Istanbul 34865, Turkey; mesut.cardio@gmail.com; 3Department of Gastroenterology, Gaziantep University Faculty of Medicine, Gaziantep 27310, Turkey; sezginbarutcu@hotmail.com; 4Department of Cardiology, Kirikkale Yuksek Ihtisas Hospital, Kirikkale 27310, Turkey; ihinanc@yahoo.com; 5Department of Cardiology, Phoenixville Hospital—Tower Health, Phoenixville, PA 19460, USA

**Keywords:** non-alcoholic fatty liver disease, electrocardiography, Tp-e interval, Tp-e/QT ratio, Tp-e/QTc ratio

## Abstract

*Background and Objectives:* Non-alcoholic fatty liver disease (NAFLD) is a common chronic liver disease associated with significant morbidity, including cardiovascular complications. This study investigates the relationship between NAFLD and electrocardiographic parameters indicative of ventricular arrhythmia risk. *Materials and Methods:* We conducted a cross-sectional study enrolling 136 patients with NAFLD and 136 healthy controls. Electrocardiographic parameters—Tp-e interval, QT and corrected QT (QTc) intervals, and Tp-e/QTc ratio—were measured and compared between groups. *Results:* Patients with NAFLD exhibited significantly higher Tp-e, QTc, Tp-e/QT ratio, and Tp-e/QTc ratio (*p* < 0.001, for all) than controls. Subgroup analysis showed progressive increases in Tp-e and Tp-e/QT ratio correlating with NAFLD severity (*p* < 0.001 and *p* = 0.001, respectively, for grade 1 vs. grade 2; *p* < 0.001 and *p* = 0.001, respectively, for grade 1 vs. grade 3). ROC analysis indicated that the Tp-e interval was a strong predictor for identifying grade 2 or more NAFLD (AUC 0.887, *p* < 0.001). *Conclusions:* Our findings highlight the association of NAFLD with prolonged electrocardiographic intervals that may predispose patients to ventricular arrhythmias. These parameters can serve as valuable markers for cardiac risk stratification in patients with NAFLD, suggesting the need for vigilant cardiac follow-up in this population.

## 1. Introduction

Non-alcoholic fatty liver disease (NAFLD) is the most prevalent chronic liver disease (CLD) in developed countries, affecting roughly 25–50% of adults [1]. NAFLD is characterized by evidence of excessive lipid accumulation in the liver, as determined by either imaging or histological examination, in the absence of considerable alcohol intake and is typically linked to the metabolic syndrome [2]. About 20% of people with NAFLD may progress to steatohepatitis, leading to liver damage that can result in fibrosis and eventually progress to cirrhosis or hepatocellular carcinoma. Beyond these potential poor outcomes, the substantial association of NAFLD with serious chronic conditions such as cardiovascular disease (CVD), chronic kidney disease, and type 2 diabetes highlights its role as a critical component of prevalent multisystem diseases in the contemporary world.

NAFLD is recognized as an independent risk factor for CVD, with CVD-related complications representing the main cause of mortality among patients with NAFLD [3]. Emerging research highlights a close association between NAFLD and the risk of cardiac arrhythmias, regardless of other conventional cardiometabolic conditions [4]. While the majority of research has focused on the association of NAFLD with atrial fibrillation, there is a notable lack of data regarding its association with ventricular arrhythmias [5,6].

On a surface electrocardiogram (ECG), the interval between the peak and end of the T wave (Tp-e) reflects the repolarization time of myocardial cells and provides insight into the distribution of transmural repolarization. The QT interval encompasses the total duration of ventricular de- and repolarization and is frequently adjusted for heart rate (QTc) in clinical studies. QTc, along with the more recently examined Tp-e and the Tp-e/QTc ratio, has been asserted to be a practical and reliable metric for identifying patients at risk of ventricular arrhythmias and sudden cardiac death. The aim of this cross-sectional study was to assess the Tp-e interval, the Tp-e/QT ratio, and the Tp-e/QTc ratio in patients with NAFLD.

## 2. Materials and Methods

The study was conducted at Kırıkkale High Specialty Training and Research Hospital. Patients were recruited in accordance with the inclusion and exclusion criteria. The sample size was calculated using the Epi InfoTM application 5.5.5. (CDC, Atlanta, GA, USA) for iOS 14.4.1. ensuring a 95% confidence interval (CI) and 90% statistical power. The patient group consisted of 136 patients (female: 71; male: 65) who were diagnosed with NAFLD in the Gastroenterology outpatient department and 136 age and sex-matched healthy volunteers (female: 61; male: 75) were included as the control group in the study. The Local Ethics Committee of Kırıkkale University approved the study, which complied with the Declaration of Helsinki’s ethical guidelines (approval date and number: 22 February 2024 and 24/16, respectively). Informed written consent was obtained from all participants.

### 2.1. Exclusion Criteria

The exclusion criteria were individuals under 18 years of age, consuming more than 20 g/day of alcohol for men and more than 10 g/day for women, those with a history of viral hepatitis or seropositive for hepatitis B virus surface antigen or anti-hepatitis C antibody, diagnosis of cirrhosis, presence of verified cardiometabolic diseases (cardiovascular diseases and chronic renal failure), presence of autoimmune or inflammatory diseases, presence of known malignancy, ECG with atrioventricular conduction disturbances, atrial fibrillation or pace rhythm, and using antiarrhythmic drugs. Other exclusions included individuals with an ejection fraction < 50%, segmental wall motion abnormalities, or moderate to severe valvular heart disease detected via routine echocardiography.

### 2.2. Clinical and Biochemical Data Collection

The clinical evaluation included taking a detailed anamnesis and physical examination. Each participant was asked to complete a questionnaire about age, sex, cigarette smoking, alcohol consumption, and medications. Smoking status was classified as a current smoker or a nonsmoker. Blood samples were drawn after at least 8 h of fasting and all biochemical analyses were performed using a parallel, multichannel analyzer (Hitachi 7600-110; Hitachi, Ltd., Tokyo, Japan) at a certified laboratory. The glomerular filtration rate (GFR) was calculated by using the CKD-EPI study equation as follows: GFR = 141 × min(Scr/κ,1)α × max(Scr/κ, 1) − 1.209 × 0.993 Age × 1.018 [if female] × 1.159 [if black], where Scr is serum creatinine (mg/dL); κ is 0.7 for females and 0.9 for males; α is −0.329 for females and −0.411 for males; min indicates the minimum of Scr/κ or 1; and max indicates the maximum of Scr/κ or 1. Weight and height measurements were performed with the participant wearing light indoor clothes and without shoes [7]. Body mass index (BMI) was calculated by dividing the weight by the square of the height.

### 2.3. Abdominal Ultrasound Investigation

All the participants underwent abdominal ultrasonography after an overnight fast of at least 8 h. Ultrasonographic examinations were conducted by two experienced radiologists who were blinded to participants’ clinical data. A Mindray DC-8 (Mindray Medical International Limited, Shenzhen, China) and a GE Logic S8 (GE Healthcare, Chicago, IL, USA) with 3.5 MHz linear transducers were the devices used during examinations.

NAFLD was diagnosed by observing an increased echogenicity of the hepatic parenchyma relative to the right renal cortex on ultrasonographic examination [8]. The degree of NAFLD severity was categorized into mild, moderate, and severe stages as outlined by Needleman et al. [9]. Mild NAFLD was indicated by a modest increase in hepatic echogenicity, with only a slight reduction in the visibility of portal venule walls. In contrast, severe NAFLD exhibited a pronounced enhancement in hepatic echogenicity, with the visibility restricted primarily to the walls of the main portal vein and the walls of smaller portal venules no longer distinguishable. Moderate NAFLD displayed characteristics that were intermediate between the mild and severe stages. NAFLD fibrosis score was calculated by using the following formula: −1.675 + 0.037 × age (years) + 0.094 × BMI (kg/m^2^) + 1.13 × impaired fasting glucose/diabetes (yes = 1, no = 0) + 0.99 × AST/ALT ratio − 0.013 × platelet (×109/L) − 0.66 × albumin (g/dL).

### 2.4. Echocardiographic Investigation

Echocardiographic assessments were conducted using a Vivid 7 pro machine (GE, Horten, Norway, 3.5 mHz) by a single cardiologist, who was blind to the clinical data of the patients. Echocardiographic measurements were made following the American College of Cardiology and American Heart Association guideline recommendations [10].

### 2.5. Electrocardiography

Each participant underwent a standard 12-lead surface ECG (ECG-1350K, Nihon Kohden, Japan) with a 10 mm/mV amplitude and a 25 mm/s paper speed after a 5 min resting period. Measurements were performed from all precordial leads for each case, and median values were used in the analyses. All ECG calculations were performed by two experienced cardiologists who were blinded to the clinical information of the patients. The Tp-e interval was defined as the time interval between the peak and the ending of the T wave. The QT interval was measured as the duration between the beginning of the Q wave and the ending of the T wave, and the corrected QT interval (QTc) was calculated by adjusting the measured QT interval for heart rate using the Bazett Formula (QTC = QT/√RR interval). The Tp-e/QTc ratio was calculated from these measurements.

### 2.6. Statistical Analysis

Statistical analyses were performed using IBM SPSS^®^ Statistics version 27 (IBM Corp., Armonk, NY, USA). The Kolmogorov–Smirnov test and skewness and kurtosis were used to determine whether the continuous variables were distributed normally. Normally distributed continuous variables are presented as the mean ± standard deviation, and non-normally distributed continuous data are presented as the median with interquartile ranges. Categorical variables are presented as frequencies and percentages. Pearson’s chi-squared test was used to compare categorical variables. The Mann–Whitney U test was used to compare non-normal continuous variables, and the independent samples *t*-test was used to compare continuous variables that were normally distributed. A one-way analysis of variance (ANOVA) test was used to compare more than two groups, and the Šídák method was used for post hoc analyses. To find the optimal cut-off values for the Tp-e, the Tp-e/QT ratio, and the Tp-e/QTc ratio linked to the presence of grade 2 or greater NAFLD, receiver operator characteristic (ROC) curves were created. Statistical significance was defined as a *p*-value of less than 0.05.

## 3. Results

A total of 136 cases with NAFLD and 136 controls were included. The patients’ clinical and demographic details are given in Table 1. The two groups were similar by means of gender, age, smoking status, and prevalence of HT and DM, whereas the NAFLD group’s mean BMI was significantly higher compared with the control group (*p* < 0.001). Regarding the laboratory parameters, serum triglyceride, aspartate aminotransferase (AST), alanine transaminase (ALT), and international normalized ratio (INR) were higher in the patient group compared with the control group (*p* = 0.006, *p* < 0.001, *p* < 0.001, and *p* = 0.014, respectively). Between the two groups, there was no statistically significant difference for the remaining laboratory parameters assessed.

Regarding ECG parameters, the NAFLD group had a significantly higher Tp-e interval, QT, QTc, Tp-e/QT ratio, and Tp-e/QTc ratio compared with the control group (*p* < 0.001 for all except *p* = 0.004 for QT). However, there was no significant difference in terms of ejection fraction, heart rate, and QRS duration between the two groups (Table 2).

Table 3 presents the ANOVA test results of the ECG parameters of the patient group, grouped according to the NAFLD grades. The subgroup analysis demonstrated that there were significant differences between NAFLD grade 1 and grade 2 and between NAFLD grade 1 and grade 3 in terms of Tp-e interval (*p* < 0.001 for grade 1 vs. grade 2 and grade 1 vs. grade 3 and *p* = 0.310 for grade 2 vs. grade 3), Tp-e/QT ratio (*p* = 0.001 for grade 1 vs. grade 2 and grade 1 vs. grade 3, *p* = 0.739 for grade 2 vs. grade 3), and Tp-e/QTc ratio (*p* = 0.003 for grade 1 vs. grade 2, *p* = 0.004 for grade 1 vs. grade 3, and *p* = 0.850 for grade 2 vs. grade 3). A statistically significant difference was observed only between NAFLD grade 1 and 3 in terms of QT duration (*p* = 0.219 for grade 1 vs. grade 2, *p* = 0.004 for grade 1 vs. grade 3, and *p* = 0.152 for grade 2 vs. grade 3) and QTc duration (*p* = 0.313 for grade 1 vs. grade 2, *p* = 0.003 for grade 1 vs. grade 3, and *p* = 0.086 for grade 2 vs. grade 3). Box-plots of Tp-e (A), Tp-e/QT (B), and Tp-e/QTc (C) indices according to NAFLD stages are shown in Figure 1.

In the ROC analysis of Tp-e interval, Tp-e/QT ratio, and Tp-e/QTc ratio designed to estimate grade 2 or more NAFLD, the area under the curve (AUC) values were 0.887 (95% CI: 0.853–0.921; *p* < 0.001), 0.730 (95% CI: 0.680–0.780; *p* < 0.001), and 0.670 (95% CI: 0.614–0.726; *p* < 0.001), respectively (Figure 2).

## 4. Discussion

In this study, we found that patients with NAFLD exhibited significantly prolonged Tp-e and QTc intervals and a higher Tp-e/QTc ratio compared with the control group. We also found that higher values of QT, QTc, Tp-e, the Tp-e/QT ratio, and the Tp-e/QTc ratio were associated with advanced-stage NAFLD. These findings suggest a potential link between NAFLD and an increased risk of ventricular arrhythmias, highlighting the importance of cardiac monitoring in this patient population.

NAFLD is a significant clinical concern due to its potential progression to non-alcoholic steatohepatitis (NASH), which is common in advanced metabolic syndrome and a leading global cause of mortality with limited treatment options [11]. NAFLD and NASH encompass a diverse population frequently burdened by cardiovascular comorbidities, including hypertension, diabetes, dyslipidemia, obesity, and atrial fibrillation. Growing evidence suggests that NASH adversely impacts cardiac function, even in the absence of these comorbid conditions [12]. Kucsera et al. experimentally induced NASH in mice via a choline-deficient, l amino acid-defined diet, observing cardiac dysfunction without the influence of obesity or metabolic syndrome. Echocardiographic and histological assessments revealed systolic and diastolic impairments, fibrosis, and increased intracardiac macrophages, further supporting NASH’s role in cardiac remodeling [12]. Our study aimed to explore whether ECG markers predictive of ventricular arrhythmias can serve as early indicators in NAFLD prior to the onset of steatohepatitis.

Increasing evidence highlights NAFLD’s association with ventricular arrhythmias. Hung et al. showed that mild to severe NAFLD correlates with prolonged QTc intervals, even after adjusting for other cardiovascular risks [13], while Mantovani et al. found a threefold increase in ventricular arrhythmia risk among NAFLD patients, independent of cardiovascular comorbidities [14].

Due to the clinical challenges associated with measuring ventricular arrhythmia incidence, Tp-e and its derivatives (the Tp-e/QT and Tp-e/QTc ratios) are valuable indicators of global ventricular repolarization abnormalities and arrhythmia risk. These markers are used across various conditions affecting cardiac electrophysiology, such as sleep apnea, acute coronary syndrome, and hypertrophic cardiomyopathy [15,16,17,18,19]. Prior studies primarily focused on the QTc interval in NAFLD. For instance, Targher et al. found that NAFLD severity correlates with prolonged QTc intervals in diabetic patients [20], while Hung et al. reported similar findings across a broad population cohort [13]. To date, only one study has examined Tp-e interval and Tp-e/QT ratio in predicting arrhythmias in NAFLD (Tak et al. observed these values to be prolonged in NAFLD patients) [21]. Our study, which included 272 participants, confirmed that QT, QTc, Tp-e, Tp-e/QT, and Tp-e/QTc values are significantly elevated in NAFLD patients, marking our study as the largest on this topic.

Several mechanisms may contribute to arrhythmogenesis in NAFLD. Cardiac and pericardial adipose accumulation produce pro-inflammatory cytokines (e.g., interleukin-1, interleukin-6, and tumor necrosis factor alpha) and reduce anti-inflammatory adiponectin, promoting arrhythmogenicity [22]. High free fatty acid levels in NAFLD accumulate as triglycerides in the heart and subsequent apoptosis of myocardial cells, leading to myocardial lipotoxicity and potential arrhythmic risk [23]. The association with cardiac autonomic dysfunction may contribute to the arrhythmia risk in patients with NAFLD [24]. Additionally, insulin resistance and intestinal dysbiosis may induce structural and electrical changes in the heart [25]. However, further investigations into the pathophysiological mechanisms could provide valuable insights for risk stratification and management in patients with NAFLD.

Its cross-sectional design, which precludes establishing causality, and the relatively small sample size are the main limitations of our study. Another major limitation of the study is that NAFLD diagnosis was based solely on ultrasound rather than using a segmentation model that can provide up to 100% accuracy in NAFLD diagnosis and risk stratification [26]. There is also the possibility that factors such as the use of herbal supplements or medications that are not yet known to cause ECG changes have not been excluded.

## 5. Conclusions

Our findings suggest that NAFLD is associated with prolonged cardiac repolarization parameters, indicating an increased risk of ventricular arrhythmias. Given the high prevalence of NAFLD and its association with significant cardiovascular morbidity and mortality, regular cardiac assessment in these patients may be warranted. Future longitudinal studies with larger cohorts are needed to better clarify our findings’ clinical implications and develop strategies for risk mitigation in this population.

## Figures and Tables

**Figure 1 medicina-61-00766-f001:**
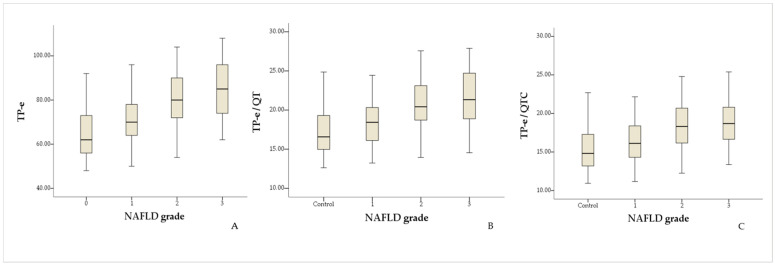
Box-plot of Tp-e (**A**), Tp-e/QT (**B**), and Tp-e/QTc (**C**) indices according to NAFLD stages. NAFLD, non-alcoholic fatty liver disease.

**Figure 2 medicina-61-00766-f002:**
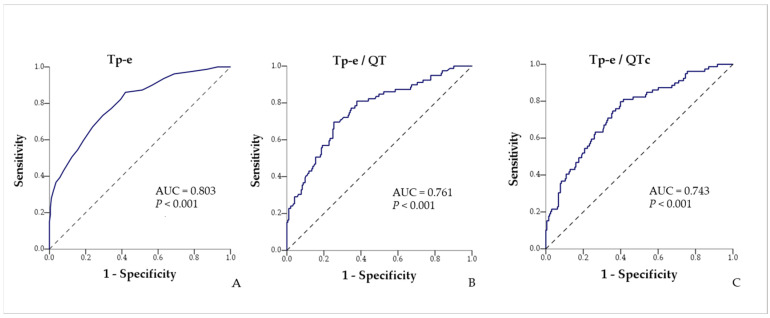
Receiver operating characteristic (ROC) curve analysis for Tp-e (**A**), Tp-e/QT (**B**), and Tp-e/QTc (**C**) indices according to NAFLD stages. AUC, area under the curve.

**Table 1 medicina-61-00766-t001:** Demographic and laboratory findings of the patient and the control groups.

VARIABLE	NAFLD (*n* = 136)	CONTROL (*n* = 136)	*p* Value
Demographic and clinical parameters			
Age (years)	50.24 ± 10.85	52.01 ± 11.17	0.184 *
Female (n/(%))	71 (52.2%)	61 (44.9%)	0.225 ^‡^
BMI (kg/m^2^)	29.54 ± 4.62	24.32 ± 3.23	<0.001 *
Smoker (n/(%))	25 (18.4%)	29 (21.3%)	0.543 ^‡^
DM (n/(%))	42 (30.9%)	31 (22.8%)	0.132 ^‡^
HT (n/(%))	35 (25.7%)	31 (22.8%)	0.572 ^‡^
Laboratory parameters			
Fasting Blood Glucose (mg/dL)	99 (76–185)	97 (71–178)	0.245 ^†^
Total Cholesterol (mg/dL)	194.92 ± 38.50	188.46 ± 35.56	0.152 *
Triglyceride (mg/dL)	182.77 ± 102.35	153.29 ± 69.38	0.006 *
LDL (mg/dL)	112.45 ± 36.64	109.99 ± 28.82	0.538 *
HDL (mg/dL)	45.93 ± 9.97	47.81 ± 10.12	0.123 *
AST (U/L)	24 (12–84)	17 (8–47)	<0.001 ^†^
ALT (U/L)	24 (10–158)	18 (6–97)	<0.001 ^†^
Bilirubin total	0.59 (0.23–2.05)	0.57 (0.17–1.33)	0.432 ^†^
Bilirubin direct	0.14 (0.05–0.58)	0.13 (0.03–0.43)	0.233 ^†^
ALP (U/L)	74.95 ± 21.69	71.24 ± 17.98	0.125 *
GGT (U/L)	23 (9–101)	24 (9–62)	0.880 ^†^
LDH (mg/dL)	190.66 ± 36.98	184.24 ± 28.83	0.112 *
INR	1.02 ± 0.11	0.98 ± 0.13	0.014 *
Creatinine (mg/dL)	0.84 ± 0.19	0.88 ± 0.176	0.131 *
GFR (mL/min/1.73 m^2^)	86.88 ± 19.47	85.33 ± 23.44	0.554 *
Total protein (mg/dL)	74.37 ± 4.81	75.24 ± 4.39	0.118 *
Albumin (mg/dL)	42.17 ± 2.97	42.18 ± 2.70	0.983 *
Sodium (mmol/L)	139.04 ± 3.17	139.44 ± 3.20	0.305 *
Potassium (mmol/L)	4.37 ± 0.37	4.40 ± 0.37	0.560 *
WBC (×10^3^/L)	7.39 ± 2.03	7.51 ± 1.78	0.615 *
Neutrophil (×10^3^/L)	4.32 ± 1.31	4.41 ± 1.25	0.530 *
Lymphocyte (×10^3^/L)	2.39 ± 0.74	2.37 ± 0.66	0.796 *
Monocyte (×10^3^/L)	0.54 ± 0.15	0.55 ± 0.17	0.531 *
Hemoglobin, g/dL	13.96 ± 1.45	14.30 ± 1.50	0.065 *
Platelet (×10^3^/L)	259.45 ± 78.41	248.950 ± 67.74	0.238 *
Sedimentation	15.77 ± 11.52	13.74 ± 11.16	0.142 *
CRP (mg/dL)	4.78 ± 6.09	3.92 ± 3.64	0.160 *

ALT, Alanine aminotransferase; AST, Aspartate aminotransferase; ALP, alkaline phosphatase; BMI, body mass index; CRP, C-reactive protein; DM, diabetes mellitus; GFR, Glomerular filtration rate; HDL, high-density lipoprotein; HT, hypertension; INR, international normalized ratio; LDH, lactate dehydrogenase; LDL, low-density lipoprotein; NAFLD, non-alcoholic fatty liver disease; WBC, white blood cell count; ‡, analyzed using the chi-square test; *, mean ± SD is presented for normally distributed data, analyzed using Student’s *t*-test; †, median (minimum–maximum) values are reported for data that are not normally distributed, analyzed using the Mann–Whitney U test.

**Table 2 medicina-61-00766-t002:** Electrocardiographic parameters of the patients.

	NAFLD (*n* = 136)	CONTROL (*n* = 136)	*p* Value
Heart rate (bpm)	77.47 ± 11.63	75.57 ± 8.17	0.121
Tp-e (ms)	76.91 ± 13.34	64.99 ± 10.26	<0.001
QT (ms)	387.56 ± 22.74	379.96 ± 20.50	0.004
QTc (ms)	438.03 ± 25.04	425.38 ± 26.73	<0.001
Tp-e/QT (%)	19.87 ± 3.38	17.16 ± 2.73	<0.001
Tp-e/QTc (%)	17.61 ± 3.16	15.36 ± 2.73	<0.001
QRS (ms)	88.60 ± 11.59	87.93 ± 10.92	0.621
LVEF (%)	59.17 ± 4.43	59.39 ± 4.91	0.697

NAFLD, non-alcoholic fatty liver disease; LVEF, Left ventricular ejection fraction; QRS, Depolarization of ventricles involving Q, R, and S waves on ECG; QT, QT interval; QTc, corrected QT interval; Tp-e, T wave peak to end interval. Mean ± SD is presented for normally distributed data, analyzed using Student’s *t*-test.

**Table 3 medicina-61-00766-t003:** ANOVA test results of the ECG parameters of the patient group, grouped according to the NAFLD stages. Grade 1, 2, and 3 hepatosteatosis.

	GRADE 1 (*n*=)	GRADE 2 (*n*=)	GRADE 3 (*n*=)	P 1-2	P 1-3	P 2-3	P 1-2-3
Heart rate (bpm)	77.49 ± 11.58	77.26 ± 11.14	77.95 ± 13.43	0.999	0.998	0.994	0.973
Tp-e (ms)	70.53 ± 10.67	80.18 ± 12.96	85.00 ± 13.59	<0.001	<0.001	0.310	<0.001
QT (ms)	381.58 ± 19.99	388.88 ± 21.81	399.64 ± 27.11	0.219	0.004	0.152	0.005
QTc (ms)	431.72 ± 27.84	438.86 ± 19.37	452.23 ± 25.36	0.313	0.003	0.086	0.004
Tp-e/QT (ms)	18.49 ± 2.72	20.65 ± 3.36	21.39 ± 3.78	0.001	0.001	0.739	<0.001
Tp-e/QTc (ms)	16.41 ± 2.75	18.32 ± 3.14	18.87 ± 3.28	0.003	0.004	0.850	<0.001
QRS (ms)	87.88 ± 11.17	88.63 ± 12.05	90.41 ± 11.76	0.921	0.984	0.999	0.845
LVEF (%)	58.91 ± 4.68	59.27 ± 4.18	59.28 ± 4.54	0.95	0.99	1	0.93

LVEF, Left ventricular ejection fraction; QRS, Depolarization of ventricles involving Q, R, and S waves on ECG; QT, QT interval; QTc, corrected QT interval; Tp-e, T wave peak to end interval. Mean ± SD was analyzed by using a one-way analysis of variance (ANOVA) test, and the Šídák method was used for post hoc analyses.

## Data Availability

The raw data supporting the conclusions of this article will be made available by the authors on request.

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
