# Peer review of "Evaluation of the Electrocardiographic Tp-e, Tp-e/QT, and Tp-e/QTc Parameters in Patients with Non-Alcoholic Liver Disease"

_medicina, 2025, doi:10.3390/medicina61040766_

Round 1
Reviewer 1 Report
Comments and Suggestions for Authors
The paper has addressed a nice issue; however, I have some comments:
- NAFLD is associated with metabolic syndrome, insulin resistance, systemic inflammation, and autonomic dysfunction, all of which can influence ECG parameters.
- It’s difficult to isolate whether NAFLD itself or its comorbidities (e.g., obesity, diabetes, hypertension) are responsible for ECG abnormalities.
- Many NAFLD patients have underlying cardiovascular disease (CVD), making it hard to determine if NAFLD independently contributes to arrhythmia risk.
- Medications (e.g., beta-blockers, statins) used by NAFLD patients can alter ECG parameters, confounding results.
- Many ventricular arrhythmia risk markers (e.g., QTc interval, Tpeak-Tend interval, QRS duration, heart rate variability) show small variations, requiring large sample sizes to detect meaningful differences.
- I would suggest investigating using MRI-PDFF or FibroScan for precise liver fat quantification instead of relying solely on ultrasound. Image segmentation plays a critical role in extracting accurate and quantitative liver fat measurements from advanced imaging techniques like MRI-PDFF (Proton Density Fat Fraction) and FibroScan (Elastography). Unlike ultrasound, which provides subjective and operator-dependent assessments, segmentation allows for automated, reproducible, and precise quantification of liver fat content. The below papers can be cited while discussing this:
“Dense-PSP-UNet: A Neural Network for Fast Inference Liver Ultrasound Segmentation,” Computers in Biology and Medicine, ScienceDirect, vol. 153, pp. 106478, 2023.
- Please include the limitations of the study.
Reviewer 2 Report
Comments and Suggestions for Authors
Non-alcoholic fatty liver disease (NAFLD) is a common chronic liver disease, often associated with cardiovascular complications. Therefore, the study of the relationship between NAFLD and electrocardiographic parameters indicating the risk of ventricular arrhythmia is relevant and timely.
The authors conducted a cross-sectional study of 136 patients with NAFLD and 136 healthy individuals (control) of the control group. The results of clinical and biochemical studies, ultrasound, echocardiographic and electrocardiographic studies were obtained, and a qualitative statistical analysis was performed. The authors showed that the differences in biochemical parameters consist of higher levels of ALT, AST and INR, as well as BMI. Regarding ECG parameters, the NAFLD group had significantly higher Tp-e interval, QT, QTc, Tp-e/QT, and Tp-e/QTc ratio compared to the control group. The results of the conducted analysis allowed the authors to discover an association of NAFLD with prolonged electrocardiographic intervals that may predispose patients to ventricular arrhythmias. These parameters can serve as valuable markers for cardiac risk stratification in patients with NAFLD, suggesting the need for vigilant cardiac follow-up in this population.
Thus, the conducted study is relevant, timely, the obtained data are correctly processed using statistical analysis methods, which allowed the authors to formulate conclusions.
I did not find any significant shortcomings in the article.
Round 2
Reviewer 1 Report
Comments and Suggestions for Authors
no more comments.